# A Deep-Learning-Based Method Can Detect Both Common and Rare Genetic Disorders in Fetal Ultrasound

**DOI:** 10.3390/biomedicines11061756

**Published:** 2023-06-19

**Authors:** Jiajie Tang, Jin Han, Jiaxin Xue, Li Zhen, Xin Yang, Min Pan, Lianting Hu, Ru Li, Yuxuan Jiang, Yongling Zhang, Xiangyi Jing, Fucheng Li, Guilian Chen, Kanghui Zhang, Fanfan Zhu, Can Liao, Long Lu

**Affiliations:** 1School of Information Management, Wuhan University, Wuhan 430072, China; 2Prenatal Diagnosis Center/Clinical Data Center, Guangzhou Women and Children’s Medical Center, Guangzhou Medical University, Guangzhou 510623, China; 3Obstetrics and Gynecology Medical Center, Dongguan Kanghua Hospital, Dongguan 523080, China; 4Center for Healthcare Big Data Research, The Big Data Institute, Wuhan University, Wuhan 430072, China; 5Medical Big Data Center, Guangdong Provincial People’s Hospital, Guangzhou 510317, China; 6Guangdong Cardiovascular Institute, Guangdong Provincial People’s Hospital, Guangzhou 510317, China; 7School of Public Health, Wuhan University, Wuhan 430072, China

**Keywords:** deep learning, artificial intelligence, genetic diseases, prenatal diagnosis, fetal face, ultrasound image

## Abstract

A global survey indicates that genetic syndromes affect approximately 8% of the population, but most genetic diagnoses can only be performed after babies are born. Abnormal facial characteristics have been identified in various genetic diseases; however, current facial identification technologies cannot be applied to prenatal diagnosis. We developed Pgds-ResNet, a fully automated prenatal screening algorithm based on deep neural networks, to detect high-risk fetuses affected by a variety of genetic diseases. In screening for Trisomy 21, Trisomy 18, Trisomy 13, and rare genetic diseases, Pgds-ResNet achieved sensitivities of 0.83, 0.92, 0.75, and 0.96, and specificities of 0.94, 0.93, 0.95, and 0.92, respectively. As shown in heatmaps, the abnormalities detected by Pgds-ResNet are consistent with clinical reports. In a comparative experiment, the performance of Pgds-ResNet is comparable to that of experienced sonographers. This fetal genetic screening technology offers an opportunity for early risk assessment and presents a non-invasive, affordable, and complementary method to identify high-risk fetuses affected by genetic diseases. Additionally, it has the capability to screen for certain rare genetic conditions, thereby enhancing the clinic’s detection rate.

## 1. Introduction

Genetic diseases, which make up around 80% of rare diseases, are caused by variations in the genome. They can result in disabilities, deformities, and intellectual disabilities in patients, and in severe cases, they can even lead to the death of children. According to a global survey, genetic syndromes affect approximately 8% of the population [1]. More than half of them impacted multiple human body systems, posing a significant burden on society [2,3]. Despite the fact that medical institutions have been using prenatal genetic technologies to screen for affected individuals, 51–89% of genetic diagnoses in the United States are made after birth [2]. According to the official documents of the Chinese government, published in 2018, the overall estimated neonatal defect rate is 5.6% [3]. Therefore, the ability to identify genetic diseases in the fetus may allow for life-saving interventions to be initiated either prenatally or early in the postnatal period.

Prenatal screening has been used to assess the risk of a fetus affected with genetic diseases since the 1970s, with the initial focus on Trisomy 21 [4]. Maternal serum assays and maternal plasma fetal cell-free fetal DNA (cffDNA) have been used to detect aneuploidy and certain types of microdeletions, such as 22q11 deletion syndrome [5]. Expanded carrier screening (ECS) has been increasingly used in recent years to reduce the risk of having a child affected by genetic diseases [6]. Nevertheless, the current costs of cffDNA and ECS are not as budget-friendly as the first-tier prenatal tests. Additionally, apart from the ECS, there have been no successful developments in effectively screening for fetal monogenetic syndrome so far. Thus, prenatal diagnosis of genetic diseases presents a global challenge, particularly in middle- or low-income countries and underdeveloped regions [7]. As a result, developing a low-cost, deployable prenatal screening strategy and diagnostic method is imperative.

Ultrasonography is a low-cost, real-time, non-invasive technique for malformation diagnosis and is widely used in prenatal screening [8]. With the development of ultrasound technology, high-quality images of the fetus can be obtained with ultrasound equipment. Prenatal screening of some diseases can be performed by a professional sonographer, and abnormalities can be detected and identified through multidisciplinary collaboration between obstetricians and geneticists. However, this process depends to a large extent on a doctor’s experience and the available equipment.

There has been significant progress in the development of deep-learning-based artificial intelligence (AI) algorithms for aiding in prenatal diagnosis [9,10], especially for structural deformity screening [11,12,13]. There are high expectations of AI applications for innovative healthcare solutions [14]. In the analysis of facial images for genetic diseases, Loos et al., first utilized facial recognition technology to diagnose five syndromes, including Cornelia de Lange syndrome (CdLS), achieving a high accuracy rate [15]. In 2016, Basel-Vanagaite et al., developed a facial recognition system called facial dysmorphology novel analysis (FDNA) technology and used it to identify facial images of patients with genetically diagnosed CdLS [16]. In 2017, Lumaka et al., conducted screening for Down syndrome patients using the Face2Gene software based on FDNA technology [17]. In the analysis of fetal facial images, Yasunari et al., successfully developed an artificial intelligence classifier to identify fetal facial expressions related to fetal brain development [18]. In 2022, the team further analyzed fetal brain activity by recognizing fetal expressions [19]. Valentine et al., used computer technology to analyze fetal facial dysmorphology and identify fetal alcohol spectrum disorders [20]. These studies demonstrate the feasibility of analyzing fetal facial information using artificial intelligence technology. However, to our knowledge, there are very few or no fetal facial analysis models developed specifically for common genetic diseases such as Down syndrome.

In this research, we developed Pgds-ResNet, a fully automated prenatal screening algorithm based on deep neural networks, to detect a variety of genetic diseases, especially some rare genetic diseases. The Grad-CAM visualization technique [21] was then used to highlight the abnormal point regions of the fetal forehead, nasal area, mouth, lip, and jaw on fetal ultrasound images. In a comparative experiment, Pgds-ResNet performed on par with senior sonographers. Our study provided an objective method for analyzing the relationship between fetal ultrasound index and genetic diseases. It also presents a solution that can assist obstetricians and sonographers in enhancing genetic disease screening during prenatal diagnosis. This solution is particularly valuable in middle- or low-income countries and underdeveloped regions.

## 2. Materials and Methods

### 2.1. Data Acquisition

From March 2020 to October 2021, the Guangzhou Women and Children’s Medical Center in China enrolled a total of 1000 pregnant women aged 23 to 38 years who underwent prenatal diagnosis (Figure 1). All ultrasound examinations were performed prior to the conduct of the research by a team of specialists with more than five years of experience. We excluded (*n* = 333), from the initial list of 1000 cases, those without genetic results, 3D volume images, and complete clinical data. Thus, our dataset contained 556 normal pregnancies and 111 pregnancies with genetic abnormalities. This is a retrospective study with genetic test results available in all cases, and the genetic results were used as the gold standard for diagnosis. The decision to induce labor or not is up to the patient.

The static 3D volume acquisition of the fetal face was performed with a high-frequency probe (6–12 MHz) equipped by a GE Voluson 730 Expert/E10 (GE Healthcare, Chicago, IL, USA). The volumes in midsagittal view were acquired using high or max quality mode and the lowest angle sweep, which allowed the inclusion of the total facial surface from the forehead to the chin. Following the collection of the ultrasound images, the sound beam angle was tilted as much as possible to 45°, the scanning volume angle range was adjusted to 50–70°, and the high-3 to max quality mode was used to set a three-dimensional volume acquisition frame of appropriate size.

### 2.2. Data Processing

All fetal facial volume images obtained were analyzed offline, using 4D View software (GE Medical Systems, Version 7.0) and activating the Volume Contrast Imaging function to optimize the volume contrast resolution. This study utilized selected 3D ultrasound volume data, ensuring that the fetal head occupied 3/4 of the screen image by adjusting the magnification. Subsequently, the image was rotated by 7° and 14° to capture the fetal facial information, with the sagittal position serving as the initial plane of interception. An automated preprocessing workflow was used to remove the identification of the information and eliminate unintended human labels. Each ultrasonic image was cropped and masked to remove text, electrocardiogram and respirometer information, and other information outside the scanning sector.

Overall, our dataset contained 845 normal images and 275 genetic disease images (including 12 genetic diseases) from 1000 pregnancies (Figure 1), with gestational week ranges from 11 to 27 years (14.8 ± 2.6). The violin diagram of gestational week values is shown in Appendix A. Finally, a program script divided all images into training and test sets. Importantly, the division of this dataset occurred at the case level rather than at the image level.

This study also employed data augmentation techniques to enhance the feature representation of images and prevent overfitting [22]. Our experiments use distortion, zoom-in, tilt, zoom-out, crop, and a combination of multiple methods to augment the training data set (Appendix A). Considering the difference in the amount of data between positive and negative samples, 2× data augmentation was used for normal images, and 6× data augmentation was used for genetic disease images. Finally, 3246 images were obtained for the training and test set samples, including 1690 normal and 1650 abnormal images.

### 2.3. Development of Deep Learning Models

Specific screening models were obtained using the fetal ultrasound images. The ResNet-18, ResNet-34, VGG-19, and VGG-16 models were trained to choose the appropriate network architecture. They are widely used in medical image analysis, especially for facial recognition [23,24]. ResNet can effectively address the vanishing gradient problem during training, allowing the network to learn deeper and more complex features [25]. The network structure of the present study consisted of the following two parts: (1) a screening model to assess the risk of fetal genetic diseases and (2) the model prediction focus areas, which were located by overlaying the heatmap and the original image.

The ResNet network model was obtained with 18 and 34 layers and the VGG model with 16 and 19 layers as the deep neural network model of this experiment. Initially, the training set was used following data enhancement to train the four aforementioned models, and the initial learning rate was set to 0.0001. Because there have been no relevant studies or models available for transfer learning or parameter initialization before, this study selected the random initialization method. The convolution kernel was set to (3,3), and Adam was employed as an optimizer. Subsequently, ReLU was selected as the activation function, and we applied binary cross-entropy to the output of the last dense layer, and the batch size was set to 32. The batch normalization module was added to each layer of the model, and weight decay (L2 regularization) was applied by us to avoid model overfitting. Subsequently, the epoch was set to 25, 50, 100, 150, and 200, and 5 models for each network structure were saved. The model with the smallest verification set loss was selected as the optimal model of the network structure.

All the functionality, experiments, and analyses were implemented using Python (NumPy 1.16 for array manipulation; opencv-python 4.1.0 and Pillow 6.0 for image operations; and scikit-learn 0.19.1 for performance quantification) and Google TensorFlow (for the implementation of the deep learning architecture).

### 2.4. Statistical Analysis

The original patient data were divided into a training set and a test set in the experiments. TensorFlow, Keras, and Python were used for statistical analysis. The performance of the genetic diseases screening model was evaluated by calculating the area under the receiver operating characteristic curve (AUROC), sensitivity, specificity, and F1*-score on the test set. The 95% CIs were the Wald CIs for sensitivity, specificity, and AUROCs, which were calculated with empirical bootstrap containing 1000 replicates. In addition, by analyzing the output results of the model on the training set, the highest F1-value was used as the selection criterion for the best threshold value.

Furthermore, receiver operating characteristic curves (ROCs) were plotted to demonstrate the screening performances and the identification of genetic diseases using the Pgds-ResNet models. The ROCs were drawn by plotting the sensitivity against the 1–specificity at different operating thresholds. Additionally, ROC analysis was performed to determine the optimal operating thresholds by using the outputs of the models on the tuning dataset. To further evaluate the performance of the deep learning model, we also used ANOVA to record the significant difference between the performance of the deep learning model and the gold standard.

### 2.5. Visualization of Facial Features in Heat Maps

The Grad-CAM visual interpretation technique was employed to generate a heatmap. This heatmap highlights the areas within the images that have a significant impact on the final classification judgment, thereby enhancing the interpretability of the model algorithm. By heatmap visualization, we were able to observe the contribution of different areas of the image to the final prediction and establish potential associations between facial features and genetic diseases.

The facial features of various genetic diseases are not clear. In order to make the prediction effect of Pgds-ResNet more convincing, the heatmap was compared with the ultrasound image.

### 2.6. Competition between Humans and AI

Three levels (junior, attending, and senior) of seniority sonographers participated in a comparison study. Initially, Pgds-ResNet and each sonographer received 100 images from the test set at random and determined whether the fetus exhibited certain genetic abnormalities independently. Secondly, the statistical analysis compared the diagnostic results with the genetic counseling report. We gave instructions to the sonographers that the examination time for each image was within 10 min. The sonographer was allowed to take a break for every 20 images examined.

The accuracy, sensitivity, and specificity were used to compare Pgds-ResNet’s screening performance with those of the human doctors. Apart from the images, the doctors were masked from all information provided.

## 3. Results

### 3.1. The Overall Framework of Pgds-ResNet

The overview of the acquisition and pretraining processing of datasets workflow is shown in Figure 1. A total of 1120 images (including 845 normal and 275 abnormal images) recorded from March 2020 to October 2021 were retrieved for developing the deep learning model named Pgds-ResNet. The case group included original images consisting of 60 for Trisomy 21 (60/275, 21.8%); 85 for Trisomy 18 syndrome (85/275, 30.9%); 45 for Trisomy 13 syndrome (45/275, 16.4%); and 85 for other genetic syndromes (85/275, 30.9%). Additionally, the control group contained 845 images without facial deformities from normal pregnancies. Finally, a total of 970 images were used for data augmentation to obtain 3340 images for training and 186 original images for testing.

The overall framework of this research is shown in Figure 2. The test set was used to evaluate the model performance of the four network structures, including ResNet-18, ResNet-34, VGG-16, and VGG-19. The optimal model was assessed by ROC performance, and the network structure with the best effect was used to produce the heatmap. In the experiment of training deep neural networks, ResNet-18 exhibited the best classification effect on the test set. Subsequently, it was optimized (see Appendix A for details) to obtain the best model effect according to the characteristics of the data set and named Pgds-ResNet. Moreover, we input the gradient of prediction results into the final convolutional layer to produce a coarse localization map highlighting the important regions in the image. Therefore, it can visualize the classification basis of the model through deconvolution technology.

### 3.2. Pgds-ResNet Outperforms Commonly Used Deep Learning Models

Pgds-ResNet yielded the following values on the test dataset after training: 0.98, 0.89, 0.96, and 0.92, corresponding to AUROC, sensitivity, specificity, and F1, respectively (Figure 3 and Table 1). When the number of network layers was increased, the AUROC of the ResNet-34 model decreased to 0.83. Notably, the sensitivity decreased from 0.89 to 0.50 due to the increase in the model complexity. If we used the VGG network without the residual blocks for comparison, the AUROC of the VGG-16 and VGG-19 models were 0.84 and 0.92, respectively, while the sensitivity could only reach 0.41 and 0.45, which indicated significant degradation in the deep neural network. The comparative experiment results indicated Pgds-ResNet is suitable for our classification task. It is worth noting that an excessively complex structure of the convolution network will reduce the sensitivity due to the relatively small sample size of the patients. Therefore, when the network depth reaches a certain degree, the performance of the deep network structure is inferior to that of the shallower neural network 19. By contrast, the residual network has reduced the degradation problem of deep learning to a certain extent due to the addition of the residual block, which significantly outperforms the VGG networks.

The *p*-value and F-value presented in Table 1 correspond to the statistical analysis performed on the performance of the deep learning algorithms. For the Pgds-ResNet algorithm, the *p*-value is 0.658, indicating no significant difference in performance compared to the gold standard. The F-value is 0.196, suggesting a relatively small difference. In general, the results indicate that the predicted outcomes of the models are fairly close to the gold standard, suggesting good predictive performance of the models. On the other hand, the corresponding F-values for ResNet-34, VGG-16, and VGG-19 are 15.921, 30.014, and 28.125, respectively, indicating a larger difference between the performance of these algorithms and the gold standard.

### 3.3. Pgds-ResNet Is Effective in Screening Common Genetic Diseases (Trisomy 21, Trisomy 18, and Trisomy 13 Syndromes)

This study conducted separate statistical analyses for each type of genetic disease. The sensitivity, specificity, and F1-score of each type of genetic disease were obtained by defining the target category (a certain type of genetic disease) as positive and the other categories as negative. In screening for Trisomy 21, Trisomy 18, and Trisomy 13, Pgds-ResNet achieved sensitivities of 0.83, 0.92, and 0.75 and specificities of 0.94, 0.93, and 0.95, respectively. With the classified threshold at 0.16, the percentage of correctly classified images into positive cases was 0.83 (10/12) in Trisomy 21, 0.92 (12/13) in Trisomy 18, and 0.75 (12/16) in Trisomy 13. In total, two Trisomy 21 images, four Trisomy 13, and one Trisomy 18 were misclassified by Pgds-ResNet as normal fetuses (false negative classification). The misclassification is primarily due to the poor image quality of those samples (Appendix A), and the main reasons for this are fetal position and sonographer experience. Compared with Trisomy 13, Pgds-ResNet performs better in Trisomy 21 and Trisomy 18. Further information, including accuracy, sensitivity, specificity, and F1-value of screening different genetic diseases, is shown in Table 2.

### 3.4. Pgds-ResNet Detects Facial Abnormalities Consistent with Clinical Reports

During the image classification, we found Pgds-ResNet focused on abnormalities within the facial region. The most prominent areas among 86 images of genetic diseases in our test set were the areas of the jaw, flat frontal bone, and nasal bone.

The ultrasound images were used to demonstrate the screening performance of Pgds-ResNet (Figure 4). The contrast results showed that Trisomy 21 exhibited the phenotypic characteristics of absent and hypoplastic nasal bones; Trisomy 13 with cebocephaly and premaxillary agenesis; Trisomy 18 with cleft lip and jaw deformity; and Turner syndrome with jaw dysplasia.

### 3.5. Pgds-ResNet Detects Rare Genetic Diseases Often Overlooked in Clinical Practice

Pgds-ResNet also detects rare genetic diseases and achieves a surprisingly high degree of accuracy, especially in cases without apparent fetal structural abnormalities. Pgds-ResNet detected 43 of the 45 other genetic disease images in the test set. The sensitivity and F1-value were 0.96 and 0.87, respectively (Table 2). The heat map also maintained consistency with their facial features. Pgds-ResNet detected abnormal signals in the middle facial area, notably in the connecting area between the nose and eyes in 1q21.1 microdeletion syndrome, areas of the nose and mouth in 15q11q13 duplication syndrome, nasal bone, and forehead in 15q26.1–q26.3 deletion with 20p13 duplication, and forehead and mouth in 17q22 microdeletion syndrome. The abnormalities in facial features in these areas overlap with ultrasound images (Figure 4) and are similar to those postnatal cases reported in the literature. Moreover, Pgds-ResNet also showed the efficacy of screening in the cases of monogenic diseases of Pyruvate dehydrogenase E1-alpha deficiency cases with mutation of PDHA1 and Helsmoortel–Van der Aa syndrome with ADNP mutation. In summary, Pgds-ResNet can effectively identify the abnormal manifestations of rare genetic diseases, which are frequently missed during sonographers’ clinical diagnoses.

### 3.6. Pgds-ResNet’s Performance Is on Par with Senior Sonographers

We also compared Pdgs-ResNet’s performance with three sonographers (Table 3 and Figure 3). The accuracies of the junior and attending sonographers were lower compared with those of the Pgds-ResNet model in screening the images for genetic diseases, i.e., accuracies of 63% by the junior sonographers, 71% by the attending sonographers, and 93% by Pgds-ResNet. Sonographers may lack training and experience in detecting facial abnormalities because they often focus on structural abnormalities in other organs, such as the heart and limbs. The senior sonographers, who had genetics training experience, achieved the best sensitivity of 0.88 and accuracy of 0.91. Therefore, Pgds-ResNet demonstrated strong performance on par with senior sonographers.

To develop a deep learning model for predicting the brain age of preterm neonates using routine clinical brain MR images, we enrolled 281 preterm infants aged 28 to 37 weeks (Figure 1 shows the distribution of participants). This was a retrospective study in which each subject received an MRI scan of the head after birth. The holdout method was employed to randomly divide the 281 MR images into two parts, one part with 211 MR images used for training and tuning and the other part containing 70 images as a test dataset.

## 4. Discussion

We propose a deep learning model called Pgds-ResNet for detecting high-risk fetuses affected by genetic diseases, especially some rare genetic diseases. Pdgs-ResNet analyzes ultrasound images and makes decisions based on the imaging characteristics that are associated with genetic diseases. Pdgs-ResNet discovers fetal facial abnormalities as the most effective features in detecting genetic diseases. The screening accuracy is on par with the senior sonographers who received genetics training before this study. Comparing the heatmaps to the ultrasound images also confirms that Pgds-ResNet is able to correctly identify Trisomy 21, 18, and 13 syndromes and a number of rare types of genetic diseases prenatally.

Recent genetic studies have shown that facial abnormalities in patients with genetic diseases are closely related to the mutation of certain genes [26,27]. For example, *10q25.3*, *8q24*, *VAX1*, *IRF6*, and other genes are associated with cleft lip disease and affect the development of the human jaw and maxilla, resulting in abnormalities in the nasal wing, cheek, lips, and other parts of the face. *Rs287104* locus in the *KCTD15* gene is related to the morphology of the nasal tip and alar; *Rs9995821* locus in the *DCHS2* gene is related to nostril aperture; *Rs2977562* locus in the *3q21.3* gene is associated with the thickness of the upper lip; and *Rs10176525* locus in the *2q36.1* gene is related to the height of the nasal bridge [28,29,30,31].

In fact, craniofacial manifestation has aided in the screening for genetic diseases [27]. Recent studies have shown that AI-based facial analysis technologies can identify genetic syndromes with similar capabilities as those of expert clinicians [32,33,34,35]. Notably, Gurovich et al., presented the facial image analysis framework DeepGestalt using computer vision and deep learning algorithms that quantify similarities of hundreds of genetic syndromes [35]. Porras et al., developed a facial deep phenotyping technology based on deep neural networks and facial statistical shape models to screen children for genetic syndromes [33]. In the field of prenatal diagnosis, the majority of studies have focused on the recognition of standard planes and the detection of anatomical structures. Only a few studies have utilized deep learning techniques to identify facial expressions for evaluating fetal development. These studies have not fully explored the potential of deep learning in identifying genetic diseases associated with facial abnormalities in fetuses. We believe that this limitation arises from the fact that deep learning techniques require a large amount of data for training, and the low prevalence of genetic diseases results in the limited availability of accumulated data in medical institutions. Additionally, factors such as image quality and the selection of standard planes in ultrasound imaging pose challenges in the development of deep learning models.

However, these genetic disease screening and diagnosis methods cannot be applied to prenatal diagnosis because the set of key facial points commonly used in facial recognition algorithms cannot be obtained in fetal ultrasound images. Our research addressed this limitation and revealed an association between fetal facial features and various genetic diseases, especially in rare genetic diseases, which enables automated screening and identification of genetic diseases based on fetal ultrasound images.

The “fetal profile” plane is a required component for standard examination during pregnancy [36]. Micrognathia, cleft lip, cleft palate, and proboscis in some cases of Trisomy 13 caused by alobar holoprosencephaly can be diagnosed by this plane. Therefore, we select this plane as the “fetal face” for AI learning.

To better interpret Pgds-ResNet’s results and minimize the black-box effect of deep learning models, we used Grad-CAM’s visualization technique to highlight the identified abnormal regions. Pgds-ResNet discovers abnormalities primarily in the areas of the facial forehead, nasal part, mouth, and jaw in different types of cases. Comparative studies on humans and AI further showed that Pgds-ResNet could discover features imperceptible to operators. Moreover, Trisomy 21, 18, and 13; Turner syndrome; 1q21.1 microdeletion; 15q11–q13 duplication; Helsmoortel–Van der Aa syndrome; 15q26.1–q26.3 deletion; and 20p13 duplication all show distinctive variations in the heatmap.

Being the first study that applies AI techniques to detect facial features in fetal ultrasound images for prenatal screening of genetic diseases, there is certainly ample room for improvement. Firstly, the limited availability of samples of fetal genetic diseases resulted in a relatively small sample size for this study. Moreover, due to the small dataset used for algorithm validation and the absence of external test sets, the results may be overly optimistic [37]. Secondly, the data utilized in this study originated from a single center, thereby constraining the applicability of deep learning models. Lastly, due to technological and theoretical constraints, the focus of this study was solely on screening for genetic diseases rather than diagnosing specific conditions. It is believed that incorporating larger and more diverse datasets in future research endeavors can enhance the robustness and generalizability of our framework, thus fostering greater advancements in the field. Moving forward, we intend to explore the potential of expanding our study to a multicenter study.

Despite these limitations, further development of our method could make a profound impact on prenatal genetic disease screening. It is well known that training a qualified sonographer is costly and time-consuming. Rapid detection and accurate diagnosis also largely depend on a clinician’s experience. A well-verified AI model with good robustness is expected to relieve the shortage of qualified sonographers, a challenge especially in under-developed regions, and hopefully reduce the financial burdens of the patients. It will also assist clinicians in making prenatal care decisions and thus improve early intervention outcomes. It can further be used as a triage scheme in clinical practice to reduce the application of NIPT or invasive procedures and save social resources. This type of screening tool could be extremely useful and is currently not available to clinicians by any other means.

In summary, this study has successfully developed an AI framework that utilizes fetal faces from ultrasound images for effective and automated screening of genetic diseases. Additionally, the framework provides informative heat maps in the context of fetal genetic conditions. Pgds-ResNet found that the fetal nose, jaw, forehead, etc., contained diagnostic information. It could help with prenatal ultrasound diagnosis, reduce false-negative results, and compensate for the lack of medical resources. However, it is worth noting that this deep-learning-based algorithm serves as an aid to doctors in diagnosis, saving them time rather than replacing them.

## Figures and Tables

**Figure 1 biomedicines-11-01756-f001:**
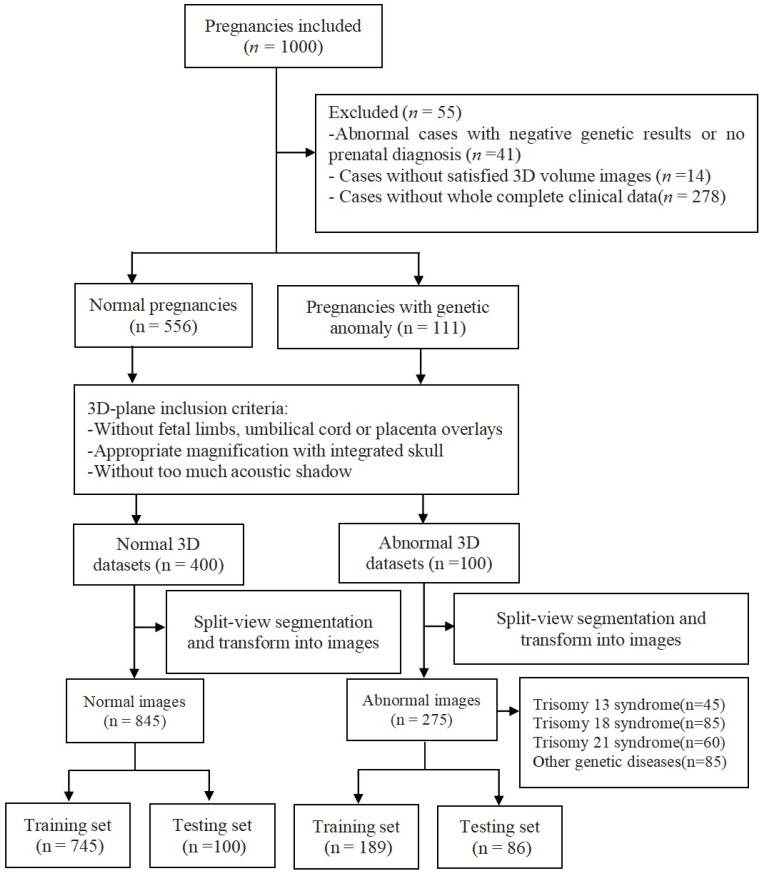
Overview of the acquisition and pretraining processing of datasets workflow. Flowchart summarizing the acquisition and pretraining processing of ultrasound images used in the training and testing sets of the deep learning algorithms for the automatic screening of genetic diseases.

**Figure 2 biomedicines-11-01756-f002:**
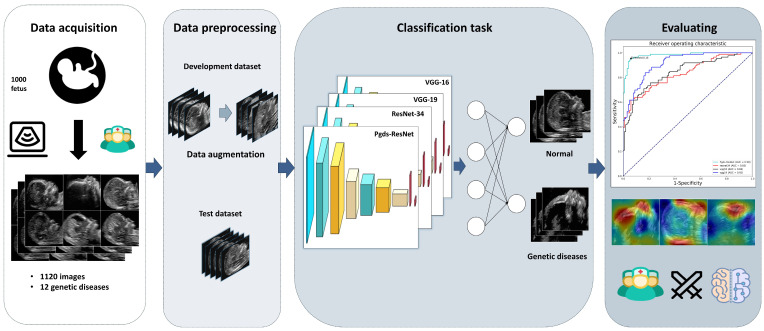
Overview of this study. Data acquisition, including clinical information and fetal ultrasound images, was performed at Guangzhou Women and Children’s Medical Center. Data preprocessing included distortion, zoom-in, tilt, zoom-out, crop, and other methods to augment the training data set. The training and testing were performed by using fetal ultrasound images to develop a deep learning model named Pgds-ResNet for the screening of genetic diseases. The model performance was assessed by AUROC, sensitivity, specificity, and F1*-score. Sonographers with three levels (junior, attending, and senior) of seniority were invited for the human–AI comparison. In case a genetic disease was detected, the abnormal areas were located by exporting the class activation mapping from the networks.

**Figure 3 biomedicines-11-01756-f003:**
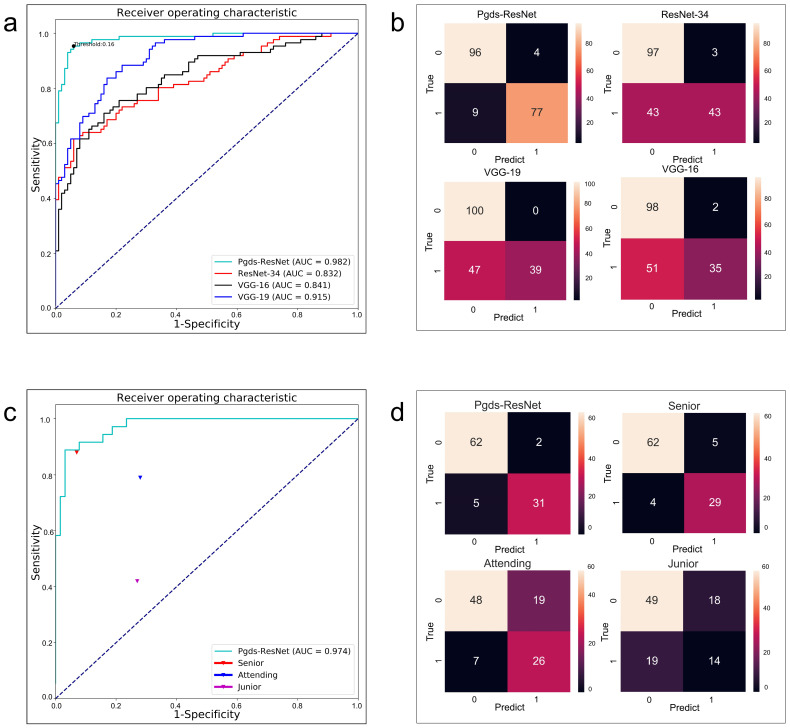
Evaluation results for Pgds-ResNet. (**a**) ROC curves for screening the presence of genetic diseases in the fetal ultrasound images. (**b**) Confusion matrix for screening the presence of genetic diseases in the fetal ultrasound images. (**c**) Comparative experiment between Pgds-ResNet and human doctors. (**d**) Confusion matrix: Pgds-ResNet framework performance compared with sonographers. AUC = area under the receiver operating characteristic curve. As shown in the color bar chart, the depth of the colors represents the quantity, with lighter colors indicating a larger quantity and darker colors indicating a smaller quantity. The dotted line represents the ROC curve of a completely random classifier.

**Figure 4 biomedicines-11-01756-f004:**
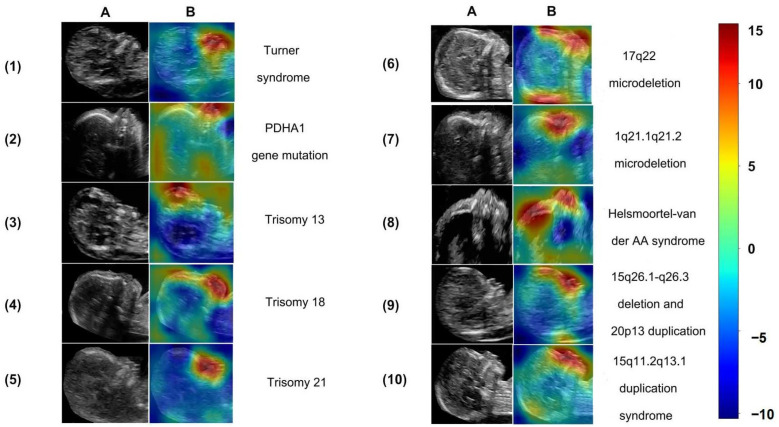
The heatmap by Grad-CAM algorithm overlaid on original images (with red regions corresponding to more attention in the heatmap on each row). The figure indicates the original images (**A**) and heatmaps from Grad-CAM (**B**). The types of diseases are as follows: (1) Turner syndrome, (2) PDHA1 gene mutation, (3) Trisomy 13, (4) Trisomy 18, (5) Trisomy 21, (6) 17q22 microdeletion, (7) 1q21.1q21.2 microdeletion, (8) Helsmoortel–Van der Aa syndrome, (9) 15q26.1–q26.3 deletion and 20p13 duplication, and (10) 15q11.2q13.1 duplication syndrome. As shown in the color bar chart, the importance values in the graph range from −10 to 15. A higher value indicates a greater importance of the pixel for the classification result. The color red represents a higher level of importance, while blue represents a lower level of importance.

**Table 1 biomedicines-11-01756-t001:** Performance of four deep learning algorithms in the test datasets.

	AUROC (95%CI)	Sensitivity (95%CI)	Specificity (95%CI)	F1	*p*-Value	F-Value
Pgds-ResNet	0.98 (0.97–0.99)	0.89 (0.80–0.95)	0.96 (0.89–0.99)	0.92	0.658	0.196
ResNet-34	0.83 (0.80–0.86)	0.50 (0.40–0.60)	0.97 (0.91–0.99)	0.65	<0.01	15.921
VGG-16	0.84 (0.81–0.86)	0.41 (0.30–0.52)	0.98 (0.92–0.99)	0.57	<0.01	30.014
VGG-19	0.92 (0.89–0.93)	0.45 (0.34–0.56)	1.00 (0.95–1.00)	0.62	<0.01	28.125

“95%CI”: 95% confidence intervals (CI) are included in brackets. “AUROC”: area under the receiver operating characteristics curve.

**Table 2 biomedicines-11-01756-t002:** Performance of Pgds-ResNet in the test datasets.

	Number	Accuracy	Sensitivity (95%CI)	Specificity (95%CI)	F1
All genetic diseases	86	0.90 (77/86)	0.89 (0.80–0.95)	0.96 (0.89–0.99)	0.92
Trisomy 13 syndrome	16	0.75 (12/16)	0.75 (0.47–0.92)	0.95 (0.90–0.97)	0.65
Trisomy 18 syndrome	13	0.92 (12/13)	0.92 (0.62–0.99)	0.93 (0.88–0.96)	0.65
Trisomy 21 syndrome	12	0.83 (10/12)	0.83 (0.51–0.97)	0.94 (0.89–0.97)	0.61
Rare genetic diseases	45	0.95 (43/45)	0.96 (0.84–0.99)	0.92 (0.86–0.96)	0.87

**Table 3 biomedicines-11-01756-t003:** The screening performance of the Pgds-ResNet model and three sonographers.

	Pgds-ResNet	Junior	Attending	Senior
Accuracy	0.93	0.63	0.74	0.91
Sensitivity	0.86	0.42	0.79	0.88
(95%CI)	(0.70–0.95)	(0.26–0.61)	(0.61–0.90)	(0.71–0.96)
Specificity	0.97	0.73	0.72	0.93
(95%CI)	(0.88–0.99)	(0.61–0.83)	(0.59–0.82)	(0.83–0.97)

95% confidence intervals (CI) are included in brackets.

## Data Availability

The data generated and/or analyzed during the current study are available upon reasonable request from the corresponding authors only for research purposes. Researchers interested in using our data must provide a summary of the research they intend to conduct. The reviews will be completed within 2 weeks, and then a decision will be sent to the applicant. The data are not publicly available due to hospital regulation restrictions. The code and example data used in this study can be accessed at GitHub (https://github.com/1057813680/Screening_genetic-diseases, accessed on 1 April 2023).

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
