# Peer review of "A Deep-Learning-Based Method Can Detect Both Common and Rare Genetic Disorders in Fetal Ultrasound"

_biomedicines, 2023, doi:10.3390/biomedicines11061756_

Round 1

Reviewer 1 Report

Overall, the idea of manuscript is average, furthermore, the contribution is not adequate at the moment. The manuscript needs significant work.  

1.      What are the limitations of the related works?

2.      Are there any limitations of this carried out study?

3.      How to select and optimize    the user-defined parameters in the proposed model?

4.      There are quite a few abbreviations are used in the manuscript. It is suggested to use a table to host all the frequently used abbreviations with their descriptions to improve the readability

5.      Explain the evaluation metrics and justify why those evaluation metrics are used?

6.      Some sentences are too long to follow, it is suggested that to break them down into short but meaningful ones to make the manuscript readable.  

7.      The title is pretty deceptive and does not address the problem completely.

8.      The related works section is very short and no benefits from it. I suggest increasing the number of studies and add a new discussion there to show the advantage. 

9.   Use Anova test to record the significant difference between performance of the proposed and existing methods.

The paper needs major revision

Author Response

Thank you for your support of our research. Below is our response to some of your suggestions.

Overall, the idea of manuscript is average, furthermore, the contribution is not adequate at the moment. The manuscript needs significant work.

Reply: We apologize if you found our manuscript lacking in innovation. We believe it may have been a misunderstanding due to our writing style. In reality, our research reveals the feasibility of utilizing deep learning techniques for genetic disease screening using fetal facial information, which has been rarely explored or even absent in previous studies. However, we acknowledge that our initial presentation of relevant research might have been insufficient, leading to your judgment. Therefore, we have added additional references to support the significance of our research, hoping to satisfy your concerns. Overall, we have made substantial improvements to the manuscript based on your requirements.

  1. What are the limitations of the related works?

Reply: There are several limitations in the existing research. In the field of prenatal diagnosis, most studies have focused on the recognition of standard planes and the detection of anatomical structures. There is limited research specifically on analyzing fetal facial images, particularly in assessing fetal development through the recognition of facial expressions. These studies have not fully explored the potential of deep learning techniques in identifying genetic diseases in fetuses that exhibit facial abnormalities. [line 379-388],[line82-85]

  1. Are there any limitations of this carried out study?

Reply:Being the first study that applies AI techniques to detect facial features in fetal ultrasound images for prenatal screening of genetic diseases, there is certainly ample room for improvement. Firstly, the limited availability of samples of fetal genetic diseases resulted in a relatively small sample size for this study. Secondly, the data utilized in this study originated from a single center, thereby constraining the applicability of deep learning models. Lastly, due to technological and theoretical constraints, the focus of this study was solely on screening for genetic diseases rather than diagnosing specific conditions. It is believed that incorporating larger and more diverse datasets in future research endeavors can enhance the robustness and generalizability of our framework, thus fostering greater advancements in the field. Moving forward, we intend to explore the potential of expanding our study to multicenter study. [line 408-420]

3.How to select and optimize the user-defined parameters in the proposed model?

Reply: hoosing and optimizing user-defined parameters in a convolutional neural network (CNN) model typically involves the following steps: [line 157-169]

More detailed:

Parameter initialization: Common initialization methods include random initialization, loading pre-trained models, and transfer learning. Because there have been no relevant studies or models available for transfer learning or parameter initialization before, this study selected the random initialization method.

Activation function selection: Popular activation functions used in CNNs include ReLU, Sigmoid, and Tanh, among others. Different activation functions can affect the model's performance and training speed. ReLU (Rectified Linear Unit) is a popular choice for an activation function in deep learning models due to several advantages it offers. Firstly, ReLU introduces non-linearity to the network, enabling it to learn complex patterns and relationships in the data. This non-linearity is crucial for capturing the intricacies of many real-world tasks. Additionally, ReLU is computationally efficient compared to other activation functions, resulting in faster training and inference times. It helps mitigate the vanishing gradient problem by maintaining a constant gradient for positive inputs, facilitating better gradient flow during backpropagation. Furthermore, ReLU can lead to sparse activation, reducing model complexity and overfitting. Overall, ReLU's ability to handle non-linear relationships, computational efficiency, and resilience against the vanishing gradient problem make it a suitable choice for a wide range of tasks, particularly in deep learning and computer vision domains. Based on this, this study selects relu as the activation function.

Learning rate adjustment: The learning rate is a crucial hyperparameter that controls the step size of parameter updates in the model. A high learning rate may cause training instability, while a low learning rate can result in slow convergence. This study selected 0.0001 as the initial learning rate and utilized the Adam optimizer to adjust the learning rate during the model training process. 

Regularization techniques: To mitigate overfitting, regularization techniques such as L1 regularization, L2 regularization, or Dropout can be employed. These techniques help control model complexity and improve generalization.

Batch size and number of iterations: Selecting suitable batch sizes and the number of iterations is crucial for training and optimization. A larger batch size can accelerate the training process but may also increase memory consumption. Taking into account the limitation of GPU memory, this study sets the batch size to 32. To observe the convergence time of the model, epochs are set to 25, 50, 100, and 150. The model with the minimum loss function during the training process is saved and considered as the optimal model.

  1. There are quite a few abbreviations are used in the manuscript. It is suggested to use a table to host all the frequently used abbreviations with their descriptions to improve the readability

Full name

Abbreviation

Expanded carrier screening

ECS

cell-free fetal DNA

cffDNA

artificial intelligence

AI

Cornelia de Lange syndrome

CdLS

Facial Dysmorphology Novel Analysis

FDNA

Area Under the Receiver Operating Characteristic

AUROC

[line 117]

  1. Explain the evaluation metrics and justify why those evaluation metrics are used?

Reply: In fact, the evaluation metrics we used are widely used in the field of deep learning, and their meanings are as follows:

Area Under the Receiver Operating Characteristic curve (AUROC): AUROC is a widely used metric in binary classification tasks. It measures the model's ability to discriminate between positive and negative instances across different decision thresholds. A higher AUROC value indicates better overall model performance and discrimination power.

Sensitivity: Sensitivity, also known as the true positive rate or recall, measures the model's ability to correctly identify positive instances. It calculates the proportion of actual positive instances that are correctly classified as positive by the model. High sensitivity indicates the model's effectiveness in identifying true positive cases.

Specificity: Specificity measures the model's ability to correctly identify negative instances. It calculates the proportion of actual negative instances that are correctly classified as negative by the model. High specificity indicates the model's ability to accurately identify true negative cases.

F1 score: The F1 score is a combined metric that considers both precision and recall. It balances the trade-off between precision (the proportion of predicted positive instances that are actually positive) and recall (same as sensitivity). The F1 score provides a single metric that reflects the model's overall performance in terms of both precision and recall.

  1. Some sentences are too long to follow, it is suggested that to break them down into short but meaningful ones to make the manuscript readable.

Reply: Thank you for your suggestion. We have taken your advice into account and made improvements accordingly. Example:[line 25-30; 35-37; 52-55; 91-95; 98-100; 192-195;……]

  1. The title is pretty deceptive and does not address the problem completely.

Reply: Thank you for your suggestion. We have made changes to the title accordingly.

 [A deep learning-based method can detect both common and rare genetic disorders in fetal ultrasound]

  1. The related works section is very short and no benefits from it. I suggest increasing the number of studies and add a new discussion there to show the advantage.

Reply: We have added relevant studies and updated the references accordingly. We have also provided a reasonable analysis and discussion based on the current state of research.

[line 67-85;379-388]

  1. Use Anova test to record the significant difference between performance of the proposed and existing methods.

Reply: In response to your suggestion, we conducted a Anova test between the predicted values of the model and the gold standard. We calculated the F-value and P-value, which indicated that the predictions of the Pgds-ResNet model were close to the gold standard. This suggests that the model exhibits good predictive performance. [line 263-271; 188-190; Table 2]

Finally, we would like to express our gratitude for your support of our research. We wish you success in your work and a joyful life!

Reviewer 2 Report

I find the article related to the application of AI in clinical practice very interesting.

On a practical level I think that generalizing in cases of rare syndromes such as 20p13, 1q21.1 microdeletion is daring. How many patients are analysed for Turner syndrome, 1q21.1 microdeletion, 15q11113 dup, Helsmoortel-van AA syndrome, 15q26.1q26.3 deletion. I don’t know if this info is in the suplementary file (I cannot open this file).

In addition, some of the clinical features of a given syndrome are not unique for a specific syndrome. For example in  Helsmoortel-van AA syndrome the characteristic facial features (prominent forehead, high anterior hairline, wide and depressed nasal bridge, and short nose with full, upturned nasal tip) described are common for other genetic conditions therefore the sensitivity and specificity of the algorithm is debatable.

On the other hand, it is true that the sonographer's time is costly and time-consuming, but the proposed algorithms will help, they will not be replaced.

Author Response

Reviewer 2

Thank you for your support of our research. Below is our response to some of your suggestions.

I find the article related to the application of AI in clinical practice very interesting.

R:Thank you for acknowledging our research. We greatly appreciate your valuable suggestions, which have broadened our perspectives for future studies.

  1. On a practical level I think that generalizing in cases of rare syndromes such as 20p13, 1q21.1 microdeletion is daring. How many patients are analysed for Turner syndrome, 1q21.1 microdeletion, 15q11113 dup, Helsmoortel-van AA syndrome, 15q26.1q26.3 deletion. I don’t know if this info is in the suplementary file (I cannot open this file). In addition, some of the clinical features of a given syndrome are not unique for a specific syndrome. For example in  Helsmoortel-van AA syndrome the characteristic facial features (prominent forehead, high anterior hairline, wide and depressed nasal bridge, and short nose with full, upturned nasal tip) described are common for other genetic conditions therefore the sensitivity and specificity of the algorithm is debatable.

R: From a practical perspective, we completely agree with your viewpoint. In this study, we generally included 1-2 cases for various rare genetic diseases. The Turner syndrome had a higher inclusion rate of 10 or more cases (this is at the case level, not the imaging level). The facial development of fetuses with rare genetic diseases is largely unknown, and much of the information we can gather comes from children after birth. However, based on the data we have acquired so far, certain rare genetic diseases do have distinctive facial features, and our team has published some case reports on this research. Overall, our approach is only to screen high-risk fetuses, and further diagnosis requires genetic testing. Therefore, the most promising application areas for this research are tiered diagnosis and large-scale screening, rather than diagnosis itself. [line 408-420]

  1. On the other hand, it is true that the sonographer's time is costly and time-consuming, but the proposed algorithms will help, they will not be replaced.

R:We hope that our manuscript has not caused any misunderstanding, and we also believe that such tools can only assist doctors in their work rather than replace them. [line 438-439]

Reviewer 3 Report

In this paper, the authors tackled an interesting and important problem of detecting abnormalities in fetal ultrasonography. The topic is certainly worthy of investigation and easily falls into the scope of the journal, and the manuscript presents interesting & valid ideas which are validated in a convincing way. There are, however, some issues which may be addressed to further improve the quality of the manuscript:

1. Please revise the table captions (it seems that the caption of Table 2 is incorrect).

2. The best results in all tables should be boldfaced.

3. It would be useful to present visual examples of correct/incorrect classification, also in relation to human readers. Such a qualitative analysis would help a reader better understand when the deep learning model introduced in this work fails.

4. The quality of the figures should be improved (all figures should be high-res). Also, please add the meaning of the color scales where appropriate (e.g., Figure 4).

5. I appreciate seeing that the authors are aware of the fact that splitting the datasets at the image level may lead to overoptimistic results, and the authors followed an appropriate approach of splitting the data at the patient level. This could be, however, discussed in more detail, and perhaps contextualized within the state of the art (see e.g., the work by Wijata for ultrasound: https://ieeexplore.ieee.org/document/9897449). 

6. Although the manuscript reads well, it would benefit from yet another pass of proofreading (see e.g., "Notably, Gurovich et al.29 presented the facial image analysis...", where the reference is incorrectly placed in the text).

Author Response

Reviewer 3

Thank you for your support of our research. Below is our response to some of your suggestions.

  1. Please revise the table captions (it seems that the caption of Table 2 is incorrect).

R: Thank you for your thorough review of our manuscript. We have made the necessary revisions. [Line 273]

  1. The best results in all tables should be boldfaced.

R: We accept your suggestion and have made the necessary corrections. [Table 2]

  1. It would be useful to present visual examples of correct/incorrect classification, also in relation to human readers. Such a qualitative analysis would help a reader better understand when the deep learning model introduced in this work fails.

R: We have included this section in Supplementary Material 4, which presents examples of misclassification and correct classification. In this study, image quality was identified as the main cause of incorrect classification results. The images in B had poor quality, which led to misclassification by the model. [Figure S4;line 567-568]

  1. The quality of the figures should be improved (all figures should be high-res). Also, please add the meaning of the color scales where appropriate (e.g., Figure 4).

R:We accept your suggestion and have provided an explanation for the color scale. The Word format may compress image quality, but we have all uploaded high-resolution images as files.[line 314-317; 296-298]

  1. I appreciate seeing that the authors are aware of the fact that splitting the datasets at the image level may lead to overoptimistic results, and the authors followed an appropriate approach of splitting the data at the patient level. This could be, however, discussed in more detail, and perhaps contextualized within the state of the art (see e.g., the work by Wijata for ultrasound: https://ieeexplore.ieee.org/document/9897449). 

R: Thank you for putting forward this suggestion, which has greatly inspired us and pointed out the direction for our future research. After reading this article, we acknowledge that our research is not rigorous enough. In future research, we will consider using standardized methods to restrict data partitioning in order to obtain more rigorous results. In the discussion, we clarified this limitation and cited relevant papers.[line 410-413]

  1. Although the manuscript reads well, it would benefit from yet another pass of proofreading (see e.g., "Notably, Gurovich et al.29 presented the facial image analysis...", where the reference is incorrectly placed in the text).

R: Thank you for your thorough review of our manuscript. We have made the necessary revisions.[line 375-377]

Round 2

Reviewer 1 Report

No further comments.

Still minor required.